## SCIENCE FORUM

# Sex differences and sex bias in human circadian and sleep physiology research

**Abstract**  Growing evidence shows that sex differences impact many facets of human biology. Here we review and discuss the impact of sex on human circadian and sleep physiology, and we uncover a data gap in the field investigating the non-visual effects of light in humans. A virtual workshop on the biomedical implications of sex differences in sleep and circadian physiology led to the following imperatives for future research: i) design research to be inclusive and accessible; ii) implement recruitment strategies that lead to a sex-balanced sample; iii) use data visualization to grasp the effect of sex; iv) implement statistical analyses that include sex as a factor and/or perform group analyses by sex, where possible; v) make participant-level data open and available to facilitate future meta-analytic efforts.

**MANUEL SPITSCHAN\*, NAYANTARA SANTHI\*, AMRITA AHLUWALIA†, DOROTHEE FISCHER†, LILIAN HUNT†, NATASHA A KARP†, FRANCIS LÉVI†, INÉS PINEDA-TORRA†, PARISA VIDAFAR† AND RHIANNON WHITE†**

**\*For correspondence:**
manuel.spitschan@tum.de (MS);
nayantara.santhi@northumbria.ac.uk (NS)

†Authors after the first two authors are listed in alphabetical order

## Introduction

Despite marked sex differences in many aspects of human physiology and behaviour, biomedical research continues to be disproportionately biased towards the male sex. For example, women made up only 25% of participants in landmark trials for congestive heart failure and 19.2% of participants for studies in antiretroviral treatment of HIV (*Criado-Perez, 2019*). Such a skewed evidence base leads to disparities in clinical and non-clinal research applications and it weakens the impact of science-based policies and translational outcomes.

This sex bias or 'sex data gap' – whereby data mainly come from male individuals – has recently received widespread attention (*Criado-Perez, 2019*), with policy advisers (*Buitendijk and Maes, 2015*), funders (*Lee, 2018*; *Clayton and Collins, 2014*) and publishers (*Rippon et al., 2017*; *Docherty et al., 2019*) pushing for better inclusivity in research regarding sex. Embracing these new practices should improve translational outcomes and scientific efficiency, but this will require a two-pronged tactic that both strengthens forces for change and weakens barriers in the field (*Karp and Reavey, 2019*).

The problems that allow sex bias to emerge are multifaceted and closing the data gap will require solutions to be bespoke for each research community.

Here, we explore the sex data gap in the context of human circadian physiology and sleep research. The field focuses on the temporal organization of physiology and behaviour at a daily scale, including rest-activity cycles, diurnal changes in hormone levels and cognitive performance, as well as the non-visual effects of light. First, we describe primary findings on sex differences in circadian physiology and sleep. Next, we discuss the sex data gap in circadian and sleep research based on an analysis of over 150 papers on the non-visual effects of light, and finally we outline recommendations emerging from a virtual workshop on the biomedical implications of sex differences in sleep and circadian physiology (held in June 2020).

While we distinguish between gender identity (how individuals and groups perceive themselves e.g. men, women, non-binary) and sex (the biological attributes that distinguish organisms as female, male or intersex), we note that these terms are often used interchangeably and wrongly

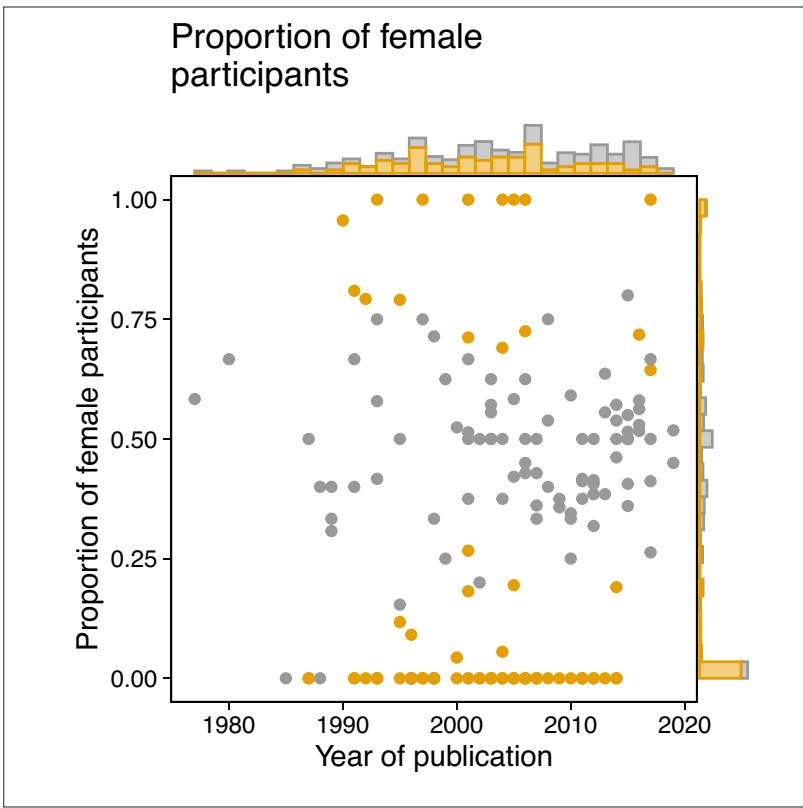

**Figure 1.** A review of the literature on the non-visual effects of light reveals a sex bias. We analyzed a sample of the existing literature on the non-visual effects of light as a starting point for understanding the sex bias in the field. The sample included a total of 180 articles, and the breakdown of participant sex was then obtained in 166 articles. Binomial tests were conducted to evaluate the possibility that deviations from an even 50:50 sex distribution were attributable to chance alone. We implemented the Benjamini-Hochberg correction for multiple comparisons to control false-discovery rate (FDR). The proportion of female volunteers in each paper (represented by a dot) was plotted against the year of publication. Samples for which the proportion of female patients deviated significantly from 0.5 ($P \le 0.05$) were determined to be biased and colour-coded as orange. The marginal histograms show the numbers of papers irrespective of publication year (histogram on the right $y$ axis), or irrespective of proportion (histogram on top $x$ axis). Methods for paper selection are included in *Methods*.

The online version of this article includes the following source data for figure 1:

**Source code 1.** R code to produce *Figure 1*.

**Source data 1.** Excel spreadsheet containing the data underlying *Figure 1*.

in the literature (*Tannenbaum et al., 2019*). Yet, in biology, sex describes differences in sexual characteristics that go beyond reproductive functions. Furthermore, we acknowledge that there is very little to no research about intersex individuals within circadian physiology and sleep research, constituting an important gap. Addressing this gap may contribute to better granularity and understanding of sex-differentiated biological mechanisms and responses. When reporting on results from the literature, we use the terms used by the researchers in these studies, as we are unable to know whether participants were asked about their sex or their gender.

## Sex differences in sleep and circadian physiology

Human circadian and sleep physiology features well-established sex differences: for instance, circadian timing is phase-advanced (earlier) in female compared to male individuals, as seen in the core body temperature minimum and evening rise in melatonin (*Boivin et al., 2016*; *Cain et al., 2010*). Female individuals also have a shorter circadian period of the temperature and melatonin rhythms (*Duffy et al., 2011*), and larger amplitude of the melatonin rhythm (*Cain et al., 2010*). Furthermore, sex differences exist in chronotype, the circadian continuum of early

('larks') to late ('owls') diurnal preference (**Chontong et al., 2016**), such that more male individuals are late types than females (**Fischer et al., 2017**; **Roenneberg et al., 2004**; **Fischer et al., 2016**; **Phillips et al., 2017**). With regard to sleep, female individuals have an earlier timing of sleep, longer sleep duration and more slow-wave sleep (**Roenneberg et al., 2004**; **Dijk et al., 1989**).

More recently, sleep regularity – the day-to-day consistency in sleep timing and duration – has emerged as an important factor in health (**Bei et al., 2016**). Irregular sleep is associated with cardiovascular disease (**Yoon et al., 2014**), inflammation (**Okun et al., 2011**), metabolic disorders (**Patel et al., 2014**; **Chontong et al., 2016**; **Spruyt et al., 2011**), mental health conditions (**Lemola et al., 2013**; **Vanderlind et al., 2014**), and cognitive impairment (**McBean and Montgomery-Downs, 2013**). The data on sex differences are mixed with reports ranging from no sex differences (**Kaufmann et al., 2016**; **Xu et al., 2018**; **Minors et al., 1998**) to more irregular sleep in female (**Mezick et al., 2009**; **Dillon et al., 2015**; **Lunsford-Avery et al., 2018**) or in male individuals (**Roane et al., 2015**; **Yetish et al., 2018**). Chronotype may account for these inconsistencies as 'owls' tend to be more irregular sleepers (**Duffy et al., 2011**; **Fischer et al., 2017**). Indeed, in revisiting three published datasets (**Fischer et al., 2016**; **Fischer et al., 2020**; **Keller et al., 2017**), more male individuals were found to be irregular sleepers than females when both were a later chronotype.

Finally, while data remain sparse, the adverse health effects of sleep irregularity itself may also differ between the sexes. To the best of our knowledge, only one study examined this question (**Roane et al., 2015**), finding that variability in sleep duration was significantly associated with weight gain in male but not female students. Overall, despite the far-reaching health implications, sex differences in sleep and circadian physiology remain underresearched.

### Impact of sex differences in sleep and circadian physiology in a non-clinical setting

Perhaps the most observable effect of sex differences in sleep and circadian physiology in a non-clinical setting is in shift work, a ubiquitous facet of modern society. Shift workers (approximately 50% of which are women) account for about a third of the workforce in North America and Europe, (**Kervezee et al., 2018**). Women have higher injury rates during night work than men, despite the injury rates between men and women being similar in day workers (**Safe Work Australia, 2009**). The physiological mechanisms underlying this difference remains unclear, partly due to a lack of research on the female circadian system. An exception is a recent study on sex differences in the effects of acute sleep deprivation on alertness (**Smolensky et al., 2017**). This work showed that women in the follicular phase of their menstrual cycle had more sleep loss-related alertness failure than men, whereas there were no differences between women in the luteal phase and men (**Vidafar et al., 2018**). This powerful influence of sex hormones and the menstrual cycle in female individuals highlight the pressing need to consider sex differences in biomedical research.

### Impact of sex differences in sleep and circadian physiology in a clinical setting

Evidence is converging that sex differences in sleep and circadian phenotypes play a role in medical conditions and should therefore be considered in medical treatments and interventions. The emerging field of chronotherapeutics or chronotherapy (**Shuboni-Mulligan et al., 2019**; **Smolensky et al., 2017**; **Dijk and Duffy, 2020**) focuses on medical treatment approaches that incorporate a patient's circadian phase, or at least the time of day, into the treatment regime. Here, we highlight a key therapeutic area, cancer treatment, in which sex-specific differences in underlying circadian mechanisms affect outcomes.

Sex and age profoundly impact chemotherapy efficacy and tolerability. Female patients are more susceptible than their male counterparts to the side effects of widely used anticancer drugs (**Milano et al., 1992**; **Stein et al., 1995**; **Chansky et al., 2005**; **Cristina et al., 2018**), and they can experience more frequent and severe toxicities from chemotherapy protocols due to sex differences in pharmacokinetics and pharmacodynamics (**Gandhi et al., 2004**). Across the 24 hour cycle, the molecular circadian clock rhythmically controls drug bioactivation, detoxification and transport while the circadian timing system as a whole regulates drug absorption, distribution, metabolism and excretion (**Dallmann et al., 2016**). In experimental models, this results in strong circadian changes in the tolerability and efficacy of over 50 anticancer medications, indicating that timing is a critical factor (**Dallmann et al., 2016**; **Lévi et al., 2010**).

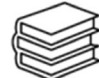

**Figure 2.** Suggested actions to close the sex data gap in sleep and circadian research for actors across the ecosystem. These actions were derived from an interactive session with attendees (n = 38) during Workshop 3.

For instance, a study examining colorectal cancer – the third cause of cancer deaths worldwide – showed that the intravenous delivery of the drug 5-FU leucovorin (5-FU-LV) at a constant rate resulted in circadian changes in drug concentration in plasma (*Petit et al., 1988*; *Fustin et al., 2012*). Most importantly, female patients had reduced 24 hour mean and circadian amplitude of the 5-FU body clearance compared to their male counterparts (*Bressolle et al., 1999*). Furthermore, peak delivery at 1pm or 4pm for oxaliplatin (another anticancer drug) and at 1am or 4am for 5-FU-LV proved to be least toxic by up to six-fold in male patients, whilst optimal timing was located six hours later in female patients (*Lévi et al., 2007*). Thus, optimal drug timing and optimal drug doses can differ according to sex (*Spitschan et al., 2020*). While the underlying mechanisms appear to involve sex differences in molecular clock function, their links with chronopharmacological determinants prompt further investigation.

## The sex data gap in sleep and circadian physiology

The sex data gap exists both in in vivo (*Curtis et al., 2018*) and in vitro research (*National Institute for Health and Care Excellence, 2020*) and is apparent even in diseases that predominantly affect women (*Patel et al., 2014*). Critically, the sex gap is not just restricted to inclusion at the experimental design stage: researchers frequently ignore sex as a factor in the analysis, even when males and females are included in the study (*Karp and Reavey, 2019*).

Apart from vision, light plays a critical role in regulating physiology and behaviour via its influence on the circadian system. These effects are mediated by a multi-component photoreceptor system consisting of rods, cones and intrinsically photosensitive retinal ganglion cells in the eye that transmit information to the circadian clock via the retinohypothalamic tract.

To ascertain whether there is a sex data gap in sleep and circadian physiology research, we focused on the non-visual effects of light on human physiology and behaviour – including how it suppresses melatonin and shifts the circadian system. This topical research area has applicability in lighting standards, regulations and guidelines (*CIE, 2019*; *International WELL Building Institute, 2020*), and various efforts are underway to incorporate scientific data from this research area into building recommendations. This highlights a pressing need to understand sex bias in this field.

A preliminary literature search identified 545 papers, which were evaluated against a list of exclusion criteria (see *Methods* for full details), yielding a total of 180 articles. In this specific analysis, we focused on the reported sex of participants, although in many instances the terms sex

and gender were used interchangeably. Each paper was then reviewed by a single reviewer to determine if participant sex and numbers were reported, and where possible, the proportion of female participants was calculated. Of the papers assessed, 14 (7.77%) did not give sufficient information on the sex breakdown of participants for this to be determined. In the remaining 166 articles, females comprised an average of 33.9% of the sample. Seven papers reported studying exclusively female participants, while 56 papers reported studying only males. *Figure 1* shows the proportion of female study participants as a function of the publication year, calculated from the per-sex participant sample sizes. We conducted binomial tests to investigate the possibility of deviations from a balanced distribution of sexes, finding a large proportion of studies using only male volunteers. Interestingly, for later years, there were fewer female-only studies (but also fewer studies in total). While this may represent a shift towards more sex-inclusive recruitment, it also means that large parts of the cited literature are based on imbalanced participant samples.

Next, we examined studies that exclusively involved male or female participants (N = 63). Of these, only eleven (17.5%) provided text to justify this sample choice. For studies with only male participants the justifications included female physiology being subject to confounding factors such as menstruation (N = 3), research into the other sex being unnecessary due to previously published observations (n = 3), the study involving a sex-specific condition (N = 2), not being able to recruit females with a specific genetic polymorphism (N = 1), the study being a case study (N = 1), and the study being conducted in a location (field station) with only male staff (N = 1). We found some evidence that the number of females increased over time, with publication year and proportion of female participants being correlated (r(164)=0.17, *P* = 0.02895). Interestingly, the total sample size correlates with the fraction of female participants in a given study (r(164)=0.3, *P* = 0.00008; Spearman's correlation): larger studies seem to recruit more balanced samples.

In summary, we find a sex data gap in the literature on the non-visual effects of light, which needs to be considered in current efforts to translate research findings in the 'real world'.

### Misconceptions underlying sex bias

One of the important aspects of an experimental design is to simplify a complex world to generate a testing space where cause and effect can be isolated. This approach is necessary to generate 'doable problems', allowing researchers to better understand the mechanisms that underlie a biologically intricate world (*Gompers, 2018*). In animal research, this simplification has led to studying one sex and strain in one batch, an approach supported by an interpretation of the 'Reduce' element of the 3R ethical framework. Historically, this has been conceived as a requirement for minimizing the number of animals in a single experiment, thus encouraging researchers to generate a narrow testing space before extrapolating and generalizing the results. Male animals were consistently selected due to the belief that the sex hormone cycle in females would lead to greater variability in the data, which would then require a larger number of female animals to achieve the same statistical power (*Karp and Reavey, 2019*). A recent meta-analysis looking at 9,932 traits found that the variability seen in female mice was not greater than for male mice – and in some cases was less – yet the legacy remains (*Prendergast et al., 2014*).

A related misconception is that studying both males and females requires the sample size to be doubled. Indeed, the analysis is not conducted independently for each sex; rather a regression analysis is used to explore the variation in the outcome variable of interest after accounting for effect of sex. Another benefit is that this approach also includes a statistical test for whether the treatment effect depends on sex. As regression analysis does not pool the data, the variance introduced by sex is accounted for, and the sensitivity for a treatment effect is minimally impacted by the inclusion of two sexes. The statistical test for the main treatment effect reveals the average treatment effect across the two sexes, and the interaction term shows how the treatment effect differs for the two sexes. The power for the main effect will be impacted when the treatment effect goes in the opposite direction for the sexes (crossed effect) but then the power to detect an interaction will increase. Biologically, crossed effects are rare, as shown in a large study assessing the prevalence of sexual dimorphism (*Karp et al., 2017*). In these situations, the treatment effect must be estimated for each sex individually. This potential situation may appear concerning to some, but it simply provides more evidence for the need to study both sexes to avoid misunderstanding biology.

Notably, the ongoing misconceptions about including female individuals in research have become part of the implicit scientific practice,

and they are passed on to future generations of researchers. To curtail this, we point to the National Institutes of Health (NIH) guidelines which stipulate that male and female sexes should be included. Furthermore, rather than automatically powering to test for an interaction, we suggest that the average treatment effect represents both sexes, and a sex-disaggregated analysis would reveal possible large differences.

Sometimes, researchers propose studying one sex at the time, but it is important to collect data on both male and female individuals simultaneously to test how the treatment interacts with sex. If data is collected independently for the two sexes, it becomes impossible to determine whether differences in estimate emerge due to sample variation or because the effect depends on sex.

A common pushback is that other sources of variation, such as age, should be considered: why should sex be the variable that is prioritized? Conducting an experiment means simplifying a complex biological world that features many sources of variation into a testing space, before generalizing the findings to reach broader conclusions. In biomedical research, the target population will be, on average, 50% male and 50% female, and it is becoming clear that variations between male and female physiology extend beyond hormonal differences. Therefore, as a rule, sex should be the first variable to be included to significantly increase generalizability – except, as discussed in the NIH guidelines, for cases such as the study of sex-specific conditions or phenomena.

## Understanding the research landscape and identifying opportunities for change

In a three-part virtual workshop held in June 2020, the authors of this paper explored practices, barriers, and challenges in designing and executing inclusive research in circadian physiology and sleep research. All materials from the workshop, including the recordings and the programme, are available under the CC-BY license (*Fischer and Vidafar, 2020*; *Karp and Ahluwalia, 2020*; *White and Lévi, 2020*).

The workshop series comprised three 90-minute sessions held a week apart and included invited talks as well as interactive sessions. The workshop was advertised through a range of channels, including Twitter, the UK Clock Club listserv, and the personal networks of the organisers and speakers. A total of 275

participants registered for the entire workshop. Across the three workshops, between 38 and 94 attendees participated in the interactive sessions, with approximately four out of five participants being researchers (82 out of 94 in Workshop 1, 47 out of 60 in Workshop 2 and 31 out of 38 in Workshop 3).

We used the web platform Mentimeter to implement polling amongst participants as well as open-ended questions. Prior to participating in the interactive sessions, attendees were informed that their responses would be used for write-up and published as a peer-reviewed article. Attendees were free to not participate in the interactive sessions. No personal data were collected as part of the interactive Mentimeter sessions. We combined yes/no, ranking and open-ended questions throughout the interactive sessions to vary the response modality. The results discussed below were selected from the results, which can be viewed in full on the Open Science Framework page. The number of responses to individual questions varied somewhat due to dropout during the interactive session as well as a time-limited response window; the total number of responses in the participatory parts are given on the bottom right-hand corner of the Materials document.

### *Workshop 1: Understanding differences*

In the first workshop, we explored sex as a variable in research. In an interactive polling segment following this workshop, only 58% of respondents (out of 100) indicated previously analyzing data in a sex-disaggregated fashion. However, 88.1% (out of 101) agreed that sex could be a variable in their research, showing the large scope for sex-disaggregated analyses. Of note, sex was identified as just one of many characteristics contributing to individual differences in research results, alongside age chronotype, mental health status, genetics, body mass index and prior light exposure. When asked for the most pressing research questions involving individual differences, the answers ranged from developmental and lifespan factors to more fundamental research questions with no obvious individual difference angle. The video recording for Workshop 1 is available here, and the materials related to the participatory part are available here.

### *Workshop 2: Understanding impact*

The second workshop focused on understanding the real-world impact of the participants' research. In the interactive polling segment following this workshop, participants indicated

## Box 1. Example journal policy to addressing sex bias.

*Amrita Ahluwalia, Editor-in-Chief of British Journal of Pharmacology (BJP)*

In 2018, the *British Journal of Pharmacology* identified the issue of sex bias in pharmacological research as a critical area for attention with respect to the work published in the journal. This came following an internal survey of our published work coupled with recognition of the activities and actions of the National Institutes of Health, in the US, raising the profile of this important issue (*National Institutes of Health, 2020*). We discovered that in addition to a prevailing reluctance to use female individuals in experimental research, both in vivo and in vitro, there was the unsurprising omission of detail regarding the sex of the source for experimental work involving primary cell culture (*Docherty et al., 2019*).

To address these issues, we introduced a number of initiatives, including: (1) publishing a themed issue in BJP containing a number of reviews and original articles focused on sex differences in pharmacology; (2) bringing together a collection of articles from all of the journals owned by the BPS in a virtual issue focused on sex; and, most importantly, (3) the elaboration and publication of guidelines for original research published in BJP. The aim of this guidance is to ensure that sex as an experimental variable is no longer ignored in articles published in BJP, but also to provide researchers with the tools to adapt their experimental design to accommodate for sex.

A key aspiration, of course, is that both male and female subjects are used as a default design in the experimental work detailed in all manuscripts submitted to the journal, but we do not mandate this at present. Our hope is that by insisting that these issues are considered within any submitted work, we raise their profile, organically leading to change. Of course, it is the responsibility of those who work with the journal to ensure that change does indeed occur. Indeed, there are many examples where such an advisory approach with other important issues related to transparency and reproducibility appear to have failed (*Leung et al., 2018*; *Avey et al., 2016*). Yet our experience in such approaches at BJP – for instance, with our guidelines on design and analysis (*Curtis et al., 2018*) – gives us strong hope that change will take place. We plan to conduct surveys of published material annually to assess this, and we will publish the outcome of these audits.

that their research could mostly influence precision and personalized medicine, occupational timing and shift/rota planning, and guidelines for an indoor 'circadian' lighting.

When asked to identify the biggest barriers to addressing sex bias in research, research money or funding and time were the most mentioned factors, followed by guidelines and policies. This indicates a scope for funding agencies to specifically address researchers' need for funding, as well as an opportunity for institutions, funders, professional bodies, learned societies and journals to develop clear guidance (see *Box 1* for an example of a journal implementing a specific policy; and *Figure 2*). The video recording for Workshop 2 is available here, and the materials related to the participatory part are here.

### Workshop 3: Understanding change

The third workshop explored factors that would facilitate change in research. In the interactive polling segment, when asked to rank sources

for guidance on sex-difference analysis, the participants first mentioned research institutes and universities, then societies and professional bodies and finally funders and publishers.

In further exploring the role of funders, the top three priorities for participants were: (1) provision of training and guidance to incorporate sex and gender analysis; (2) allocation of funding within regular grant mechanisms ringfenced for sex and gender analysis; and (3) simply more allocation of funds in regular research grants. Additionally, collaboratively developed guides, research toolkits, training programmes from societies and professional bodies were also indicated as facilitators of change.

When asked what researchers could personally do, three actionable items emerged: (1) inclusion of sex and gender analysis as a central step in research; (2) learning from peers and with examples; and (3) upskilling in the requisite statistical techniques. The video recording for Workshop

## Box 2. Patient and Public Involvement as a vehicle to make research more inclusive.

Patient and Public Involvement (PPI) (*National Institute for Health and Care Excellence, 2020*; *Arthritis Research UK, 2017*; *Bagley et al., 2016*) is defined as research carried out 'with' or 'by' patients, those who have experience of a condition, and the broader public in general. PPI is a term that is largely used in the UK research landscape, but similar initiatives may exist in different countries. PPI differs markedly from engagement and participation; this refers to various types of interactions with people with a condition (such as providing information and knowledge in research) as well as surveying what people understand about a particular condition regardless of whether they experience it, or exploring what should be prioritized in basic or clinical research on that condition. Involvement, on the other hand, implies a more active collaboration between researchers, and the target group – and in some cases the general public – that helps shape the design of a research project. At different levels, all these interactions provide opportunities for dialogue and bring research to those directly impacted by conditions and the public. This, in turn, helps increase diversity – including, but not limited to, making research more inclusive with respect to sex and gender. Engaging with the general public and with patients is now often asked by charities and research funding organizations but should be considered beyond being a box-ticking exercise. PPI will very likely impact the design of research projects by identifying what is vital to patients and society, and why. In turn, this will help to identify gaps in our understanding of the disease or condition in question thereby increasing research quality. This can help prioritize research areas and lead to research that is better aligned with patient and public interests. For example, the James Lind Alliance Priority Setting Partnerships is a non-profit initiative bringing patients, carers and clinicians together to identify and prioritise unresolved questions or evidence uncertainties they consider important. In this way, research funders become aware of what matters most to the people who use their research in their everyday lives. PPI will also help the target group to better understand research, and give an often unique opportunity for researchers – especially discovery scientists – to understand patients' reality and perspective.

three is available here, and the materials related to the participatory part are available here.

## Recommendations: Guiding principles to close the sex data gap

Based on the workshop content and discussions, we propose the following guiding principles to address the sex data gap in biomedical research, and to build an evidence base which is better inclusive of sex and gender. The central tenet includes sex and gender analysis as an essential component of research design. The specific recommendations are:

1. Design research to be inclusive and accessible. In many cases, research is designed exclusively by researchers who may not necessarily have sufficient expertise on how to make their study inclusive and accessible. An important step is reaching clarity in recording and reporting participant sex and gender. As an example, one research

team reporting the sex of participants may use participant-derived responses on a questionnaire or intake form, and another group may use the sex assigned at birth, based, for instance, on an ID card. While these could give congruent answers, they represent different types of information. Wider engagement with definitions of sex and gender and questions surrounding this topic within a research group or researcher community could lay the groundwork for making research more inclusive and accessible. As a formalised way to ensure inclusivity, we also suggest that research participants be integrated in the research planning process through Patient and Public Involvement (see *Box 2*) or similar mechanisms.

2. Implement recruitment strategies that lead to a sex-balanced sample. This includes wide advertisement of research studies, and tailoring recruitment strategies by engaging with patients, participants and the general public, for example through Patient and Public Involvement

mechanisms (see *Box 2*). Given fixed resources, recruiting a sex-balanced sample does not simply mean doubling the sampling size, but merely recruiting a sample with 50% female and 50% male participants. A balanced design is recommended to ensure the resulting statistical analysis is robust and that the variance can be decomposed to the factors of interest without confounding these (*Collins et al., 2009*). While exceptions to this principle may arise from sex-specific research questions, as a general guiding principle there is little to argue against. Furthermore, this will allow sex to be included as a factor in the analysis without compromising sensitivity to a generalizable main effect.

3. Use data visualization to grasp the effect of sex. An informal visualization in the early stages of analyses can be used to ascertain sex difference trends, which can then be followed up with more rigorous statistical testing.

4. Implement statistical analyses that include sex as a factor and/or perform group analyses by sex, where possible. Sex can be included as a factor or a covariate in analyses, or an alternative strategy can be to perform a group analysis by sex. Both require a good understanding of effect sizes and statistical power. Researchers should seek to upskill in statistics to develop advanced analytic strategies.

5. Make participant-level data open and available to facilitate future meta-analytic efforts. This step requires data to be available, which many journals now mandate. The large, combined sample size afforded by the wide availability of data can enable a sex-related effect to be more readily detectable. We also suggest that researchers should include tables reporting the primary data and participant meta-data as supplementary information in articles. A recent analysis of open science practices in circadian rhythms and sleep research journals (*Spitschan et al., 2020*) has indicated an opportunity to mandate data sharing in journal policies. Journal policies requiring participant-level data sharing could facilitate future analyses incorporating sex.

While none of these actions will suffice on their own, each will contribute to closing the sex data gap. Of course, the research ecosystem not only includes individual researchers but also institutions of varying sizes. We present multiple actions that can be adopted by institutions, funders, as well as professional bodies, learned societies, journals in *Figure 2*. These actions were developed from an interactive segment of Workshop 3.

## Methods

To implement a breadth-first search for identifying relevant papers, we employed a pragmatic hybrid strategy, identifying relevant articles through three main sources, as listed in *Table 1*. We conducted a citation search of three key recent reviews (*Brown, 2020*; *Souman et al., 2018*; *Lok et al., 2018*) on the acute effects of light, producing a total of 88 papers of which 83 were included in the present analysis. We carried out a search for papers specifically discussing the melatonin-suppressive effects of light in SCOPUS (search carried out on 22 October 2019) through the search term "TITLE-ABS-KEY

**Table 1.** Articles included in the meta-analysis.

| Database | Search strategy | Source paper | Articles considered | Articles included |
|---|---|---|---|---|
| – | – | *Brown, 2020* | 19 | 18 |
| – | – | *Lok et al., 2018* | 20 | 20 |
| – | – | *Souman et al., 2018* | 49 | 45 |
| SCOPUS | Citation count | - | 359 | 94 |
| | | *Pachito et al., 2018* | 5 | 0 |
| | | *Forbes et al., 2014* | 13 | 0 |
| | | *Montgomery and Dennis, 2002* | 0 | 0 |
| | | *Tuunainen et al., 2004* | 49 | 3 |
| | | *Slanger et al., 2016* | 21 | 0 |
| Cochrane | (light AND circadian OR sleep OR alertness)" | *Dennis and Dowswell, 2013* | 10 | 0 |
| | | | 545 | 180 |

((light AND melatonin AND suppress*)) AND (LIMIT-TO (DOCTYPE, "ar")) AND (LIMIT-TO (LANGUAGE, "English"))" (search carried out on 22 October 2019). Limiting the analysis to papers with a minimum of 30 citations, we identified 359 further papers (94 of which were included). Finally, relevant systematic reviews were identified in the Cochrane Library through the search terms "(light AND circadian OR sleep OR alertness)", generating 24 results with six relevant for the present analysis. A citation search was again conducted, generating a further 98 papers (of which three were included). Overall, a total of 545 papers were identified and analyzed, as shown in *Table 1*.

### Inclusion and exclusion criteria

Papers were excluded where the following exclusion criteria applied, leaving a total of 180 papers for the present analysis:

1. Studies that do not assess the acute effect of light: including those looking at longitudinal exposures or habits rather than controlled light exposure within a specified time frame, e.g. cohort and case-control studies were excluded;
2. Studies in which the primary outcome measure did not relate to circadian physiology (e.g. the role of light exposure in treating affective disorders);
3. Studies assessing the effects of interventions other than light exposure, e.g. sleep deprivation or magnetic field exposure. In papers involving multiple studies, only those assessing the acute effects of light were included, with other studies excluded;
4. Studies for which the PDF of the paper could not be obtained, or could not be obtained in English;
5. Studies primarily focusing on non-human animals;
6. Review papers, opinion pieces or commentaries not including any primary data;
7. Studies not based on measurements taken from human participants, e.g. in vitro studies or mathematical models. Measurements of human materials such as blood or retinal cells were considered to be from human participants if the intervention (light exposure) was carried out before the material was isolated from participants, but they were excluded if measurements were taken after the materials were obtained;
8. Research involving participants under the age of 18;
9. Studies in which variables were not manipulated (i.e., naturalistic or observational studies);
10. Field studies, in which variables were manipulated outside of a laboratory setting.

Papers were not excluded based on participant disease status or outcome measure. No upper limit was set for participant age. In coding the articles, we did not make a distinction between sex and gender, as these are conflated in the literature.

**Manuel Spitschan** is in the Department of Sport and Health Sciences, Technical University of Munich, Munich, Germany, the Translational Sensory and Circadian Neuroscience Research Group, Max Planck Institute for Biological Cybernetics, Tübingen, Germany and the Department of Experimental Psychology, University of Oxford, Oxford, United Kingdom

manuel.spitschan@tum.de

http://orcid.org/0000-0002-8572-9268

**Nayantara Santhi** is in the Department of Psychology, Northumbria University, Newcastle-upon-Tyne, United Kingdom

nayantara.santhi@northumbria.ac.uk

http://orcid.org/0000-0003-4568-1447

**Amrita Ahluwalia** is at the William Harvey Research Institute, Barts and The London School of Medicine and Dentistry, Queen Mary University of London, London, United Kingdom

http://orcid.org/0000-0001-7626-6399

**Dorothee Fischer** is at the German Aerospace Center, Institute of Aerospace Medicine, Sleep and Human Factors Research, Köln, Germany

http://orcid.org/0000-0002-2122-3938

**Lilian Hunt** is at the Wellcome Trust, London, United Kingdom and in the Equality, Diversity and Inclusion in Science and Health Group, London, United Kingdom

http://orcid.org/0000-0002-0319-7764

**Natasha A Karp** is in the Data Sciences and Quantitative Biology division, Discovery Science at AstraZeneca R&D, Cambridge, United Kingdom

http://orcid.org/0000-0002-8404-2907

**Francis Lévi** in at the Warwick Medical School, University of Warwick, Warwick, United Kingdom, the Hepatobiliary Center, Hospital Paul Brousse (AP-HP), Villejuif, France and the UPR Chronotherapy, Cancer and Transplantation, Medical School, Paris-Saclay University, Gif-sur-Yvette, France

http://orcid.org/0000-0003-1364-7463

**Inés Pineda-Torra** is in the Centre for Cardiometabolic and Vascular Science, Division of Medicine, University College London, London, United Kingdom

http://orcid.org/0000-0002-7349-2208

**Parisa Vidafar** is in the Sleep and Circadian Research Laboratory, Department of Psychiatry, University of Michigan, Ann Arbor, United States, and the School

of Psychological Sciences and Turner Institute for Brain and Mental Health, Monash University, Clayton, Australia

http://orcid.org/0000-0002-3990-1047

**Rhiannon White** is in the Department of Experimental Psychology, University of Oxford, Oxford, and the Warwick Medical School, University of Warwick, Warwick, United Kingdom

http://orcid.org/0000-0002-8175-3586

*Author contributions:* Manuel Spitschan, Conceptualization, Funding acquisition, Investigation, Methodology, Project administration, Visualization, Writing – original draft, Writing – review and editing; Nayantara Santhi, Conceptualization, Writing – original draft, Writing – review and editing; Amrita Ahluwalia, Writing – original draft, Writing – review and editing; Dorothee Fischer, Investigation, Methodology, Writing – original draft, Writing – review and editing; Lilian Hunt, Conceptualization, Resources, Writing – original draft, Writing – review and editing; Natasha A Karp, Writing – original draft, Writing – review and editing; Francis Lévi, Writing – original draft, Writing – review and editing; Inés Pineda-Torra, Conceptualization, Writing – original draft, Writing – review and editing; Parisa Vidafar, Writing – original draft, Writing – review and editing; Rhiannon White, Investigation, Methodology, Visualization, Writing – original draft, Writing – review and editing

*Competing interests:* Amrita Ahluwalia: Editor-in-Chief for the British Journal of Pharmacology. Lilian Hunt: employee of the Wellcome Trust. Her work is unrelated to the Wellcome Trust's Research Enrichment funding mechanisms that funded the workshop. The other authors declare that no competing interests exist.

## Funding

| Funder | Grant reference number | Author |
| --- | --- | --- |
| Wellcome Trust | 204686/Z/16/A | Manuel Spitschan |
| Deutsche Forschungsgemeinschaft | FI 2275/3-1 | Dorothee Fischer |

The funders had no role in study design, data collection and interpretation, or the decision to submit the work for publication.

**Decision letter and Author response**
Decision letter https://doi.org/10.7554/eLife.65419.sa1
Author response https://doi.org/10.7554/eLife.65419.sa2

## Additional files

### Supplementary files
• Transparent reporting form

### Data availability
All data generated or analysed during this study are included in the manuscript and supporting files. Source data and source code files have been provided for Figure 1.

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
