## [Decision Letter]

**Decision letter after peer review:**

Thank you for submitting your article "Sex differences and sex bias in circadian rhythms and sleep research" to *eLife* for consideration as a Feature Article. Your article has been reviewed by three peer reviewers, and the evaluation has been overseen by a Reviewing Editor and Peter Rodgers as the Senior Editor.

The reviewers and editors have discussed the reviews and we have drafted this decision letter to help you prepare a revised submission.

Please also note that *eLife* does not permit figures in supplementary materials: instead we allow primary figures to have figure supplements (tables, however, can be supplementary materials).

After revision, please note that the *eLife* Features Editor may also contact you separately about some editorial issues that you will need to address.

Summary:

This article highlights the sex-bias present in chronobiology and sleep medicine research, and it provides recommendations and guidance for sex- and gender-inclusive research practices and policies. To do so, it relies on insight from a workshop conducted to evaluate sex differences in circadian and sleep research. Overall, the authors have provided robust examples from a growing body of evidence which suggests that biological sex influences the human circadian clock and that these sex differences may play a key role in health and disease. However, a number of points need to be addressed to make the article suitable for publication.

Essential revisions:

1. Please shorten the introduction, and reorganize the first two sections "Sex differences are ubiquitous in circadian rhythms and sleep" and "Sex differences in circadian rhythms can impact therapeutic interventions" into one cohesive introduction section which lays the ground for the literature analysis and the workshop discussion. In doing so, please make sure to streamline the text to ensure the reader feels it was written cohesively rather than by different people.

2. Lines 61-198 have a large overlap of topic matter with lines 200-266: please try to reduce this repetition. (For instance, the sub-section "What is the sex data gap?" repeats the conclusions from the previous section).

3. Line 73: Please discuss this point further, as "blockers" are not the only people who may resist change – they are probably a tiny minority of people. Far more people resisting change are probably people who don't know enough about how or why.

4. Line 217 onward: Please discuss why there appears to have been a thorough review of studies involving light impact on human circadian rhythms, but not of other types of studies that are only mentioned briefly (e.g., circadian period or phase)

5. Line 227 onwards: Please clarify whether there was a relationship between the proportion of female participants and the sample size of the study

6. Line 268 onwards: Please provide further information about the workshop participants and how they contributed to the outcomes of the paper.

– For instance, please clarify the methods used to obtain participant reflections/responses (i.e. Did you provide participants with a certain set of questions or surveys to obtain the information described in Workshops 1 and 2? Alternatively, did this information come from discussions which were transcribed and coded?)

– Please indicate whether the participants of the workshop gave consent for their responses/comments to be used in this format? If not, please discuss whether there is a need for a retrospective-IRB approval.

– Please report how many individuals attended each workshop

– For each workshop section, please give all answers, not a selection of them. You can group into categories or ones with similar wording. The number of each question is not needed since there was limited time to respond and participants may not have been able to include all possibilities (so the number of each response does not necessarily reflect the importance of said response).

7. Line 282: Please further discuss the statement that a common misconception in research is that the sample size must be doubled to include both sexes. The authors note that this is a flawed assumption because treatment effects are reported from the entire group. While this is certainly true, the authors should also acknowledge that the issue may be more complex than described, especially because sex differences exist. In the first half of this manuscript, the authors describe important differences that have been identified between sexes on a variety of outcomes (shiftwork tolerance, sleep differences, phase differences, etc.). Doesn't this suggest that pooling of males and females a bad idea, particularly for time-varying differences? For example, on lines 188-192, the authors describe the results of several important studies examining oxaliplatin and irinotecan in male and female patients. The authors note that the optimal timing for females and males were at different times for these treatments. If these reports would have pooled the data for males and females wouldn't that have attenuated the results or obscured the researchers' ability to observe a time-of-day effect at all? In this case, it seems pretty important that the analyses were stratified by sex. If a researcher wishes to examine outcomes by sex, as seems pretty important given the thesis of this manuscript, then one should be powered to conduct separate analyses (Diester CM, Banks ML, Neigh GN, Negus SS. Experimental design and analysis for consideration of sex as a biological variable. Neuropsychopharmacology. 2019;44(13):2159-2162. doi:10.1038/s41386-019-0458-9).

8. Line 343 onwards (Recommendations #2):

– Please discuss further whether 1:1 female:male is the only valid ratio. Would other ratios be ok in some circumstances? When is this ratio more important than other variables (e.g. age)?

– Please acknowledge that recruiting a sample that consists of 50% females and 50% males may increase variability (in some circumstances) compared to studies with 100% of one sex, and therefore change power calculations and/or require larger sample size.

9. Recommendations: Please consider discussing whether one option for closing the gap would be to fund/conduct studies that only include females.

10. Line 315: Please provide a copy of the workshop program booklet or guide as supplemental information, so that "the way [you] constructed these workshops and facilitated discussion around sex could be used as a template for exploring other aspects of difference in future as research progresses". The reviewer was unable to find this information based on the references provided [75-77].

11. Line 433 onwards (Methods): Please report whether one or more authors conducted the bibliometric analyses. If multiple authors contributed, please indicate whether there were any inter-rater reliability tests done to assess accurate coding.

12. Please provide the source data for the bibliometric analysis, which should contain citation data and sex-based coding as presented in Figure 1.

---

## [Author Response]

Essential revisions:1. Please shorten the introduction and reorganize the first two sections "Sex differences are ubiquitous in circadian rhythms and sleep" and "Sex differences in circadian rhythms can impact therapeutic interventions" into one cohesive introduction section which lays the ground for the literature analysis and the workshop discussion. In doing so, please make sure to streamline the text to ensure the reader feels it was written cohesively rather than by different people.

We have now shortened the introduction and combined the two sections, thereby streamlining the flow and harmonising style.

2. Lines 61-198 have a large overlap of topic matter with lines 200-266: please try to reduce this repetition. (For instance, the sub-section "What is the sex data gap?" repeats the conclusions from the previous section).

We have now revised this section and removed repetitions.

3. Line 73: Please discuss this point further, as "blockers" are not the only people who may resist change – they are probably a tiny minority of people. Far more people resisting change are probably people who don't know enough about how or why.

The term blockers was used in reference 7 (Karp et al., 2017) in a change management reflection on the resistance to studying both sexes. So the term blocker doesn’t refer to people but encompasses a myriad of practical, cultural and other issues. We have rewritten the paragraph to improve clarity:

“Sex bias or the 'sex data gap' has recently gained widespread attention (Criado-Perez, 2019), with policy advisers (Buitendijk and Maes, 2015), funders (Lee, 2018; Clayton and Collins, 2014) and publishers (Rippon et al., 2017; Docherty et al., 2019) pushing for the inclusion of both sexes in research. […] This two-pronged approach will unfreeze the status quo and allow new approaches to emerge (Karp and Reavey, 2019).”

4. Line 217 onward: Please discuss why there appears to have been a thorough review of studies involving light impact on human circadian rhythms, but not of other types of studies that are only mentioned briefly (e.g., circadian period or phase)

We use the area of non-visual effects of light as a very tangible example in this article. The area is very topical. We have now clarified this in the text:

“Because there are various efforts underway to incorporate scientific data from this research area into guidelines and recommendations for the provision of lighting in the built environment, there is a pressing need to understand sex bias in this area, which we have prioritised in the analysis reported here.”

5. Line 227 onwards: Please clarify whether there was a relationship between the proportion of female participants and the sample size of the study

We have included this additional analysis. We have added the following sentence:

“Interestingly, we also find that the total sample size correlates with the fraction of female participants in a given study (r(164) = 0.23, p = 0.00275).”

6. Line 268 onwards: Please provide further information about the workshop participants and how they contributed to the outcomes of the paper.– For instance, please clarify the methods used to obtain participant reflections/responses (i.e. Did you provide participants with a certain set of questions or surveys to obtain the information described in Workshops 1 and 2? Alternatively, did this information come from discussions which were transcribed and coded?)– Please indicate whether the participants of the workshop gave consent for their responses/comments to be used in this format? If not, please discuss whether there is a need for a retrospective-IRB approval.– Please report how many individuals attended each workshop– For each workshop section, please give all answers, not a selection of them. You can group into categories or ones with similar wording. The number of each question is not needed since there was limited time to respond and participants may not have been able to include all possibilities (so the number of each response does not necessarily reflect the importance of said response).

We have expanded upon the description of the workshops in both the workshop sections and have added a methods section describing these aspects. All workshop materials have been made available on the Open Science Framework (https://doi.org/10.17605/OSF.IO/WU2QX). We have limited the discussion of findings to key results from the interactive sessions.

We have added information regarding participant consent in the methods section:

“Prior to participating in the interactive sessions, attendees were informed that their responses will be used for write-up and published as a peer-reviewed article. […] No personal data were collected as part of the interactive Mentimeter sessions.”

We have added information about attendance numbers in the main section:

“A total of 275 individuals registered for the workshop. Across the three workshops, between 38 and 94 attendees participated in the interactive sessions, with approximately 4/5 of all participants being researchers (82/94 in Workshop 1, 47/60 in Workshop 2, 31/38 in Workshop 3). […] Responses to individual questions varied somewhat due to drop-out during the interactive session as well as a time-limited response window; the total number of responses in the participatory parts are given on the bottom right-hand corner of the materials document.”

We have added information on the question methodology as well:

“Throughout the interactive sessions, we combined yes/no, ranking and open-ended questions to vary the response modality. The results discussed below were selected from the results, which can be viewed in full on the Open Science Framework page above.”

7. Line 282: Please further discuss the statement that a common misconception in research is that the sample size must be doubled to include both sexes. The authors note that this is a flawed assumption because treatment effects are reported from the entire group. While this is certainly true, the authors should also acknowledge that the issue may be more complex than described, especially because sex differences exist. In the first half of this manuscript, the authors describe important differences that have been identified between sexes on a variety of outcomes (shiftwork tolerance, sleep differences, phase differences, etc.). Doesn't this suggest that pooling of males and females a bad idea, particularly for time-varying differences? For example, on lines 188-192, the authors describe the results of several important studies examining oxaliplatin and irinotecan in male and female patients. The authors note that the optimal timing for females and males were at different times for these treatments. If these reports would have pooled the data for males and females wouldn't that have attenuated the results or obscured the researchers' ability to observe a time-of-day effect at all? In this case, it seems pretty important that the analyses were stratified by sex. If a researcher wishes to examine outcomes by sex, as seems pretty important given the thesis of this manuscript, then one should be powered to conduct separate analyses (Diester CM, Banks ML, Neigh GN, Negus SS. Experimental design and analysis for consideration of sex as a biological variable. Neuropsychopharmacology. 2019;44(13):2159-2162. doi:10.1038/s41386-019-0458-9).

There are ways in which sex can impact an outcome variable of interest. Firstly, the variables baseline levels can depend on sex. Secondly, the treatment effect can depend on sex. When studying both sexes, a regression analysis will account for the sex differences in baseline levels and this has minimal impact on power. Regression analysis is not pooling the data which is completely inappropriate and can deliver misleading results. Recommending half the animals are male and half are female for a treatment group, is based on the premise that you want to estimate an average effect and then you can see a large interaction if it occurs. If we are interested in these differences, we then need the power to study the interaction which is a different goal.

In typical time course studies, where the time effect is also of interest, a regression analysis would ask at each time point what is the effect of sex, what is the effect of the treatment and does the effect depend on sex.

The section on page 8 has been reworded to give greater clarity and to highlight that pooling is not recommended.

8. Line 343 onwards (Recommendations #2):– Please discuss further whether 1:1 female:male is the only valid ratio.

Statistically it is good practice to have balanced designs [ref: https://www.ncbi.nlm.nih.gov/pmc/articles/PMC2796056/]. ‘In a balanced design the main effects and interactions are orthogonal so that each one is estimated and tested as if it were the only one under consideration, with very little loss of efficiency due to the presence of other factors’. Basically, having a balanced design makes analysis more robust and ensures that factors are independent and the variance can be decomposed in to the individual contributions without confounding. Consequently, this is our recommendation as the default basic design. The following sentence has been added to page 11 “A balanced design is recommended to ensure the resulting statistical analysis is robust and that the variance can be decomposed to the factors of interest without confounding.”

Would other ratios be ok in some circumstances? When is this ratio more important than other variables (e.g. age?)

We have added a paragraph to page 9:

“A common push back is that there are other sources of variation (e.g. age) that should be considered and ask why should sex be the variable that is prioritized. Whenever we conduct an experiment we are simplifying a complex biological world with lots of sources of variation into a testing space and we generalize the findings allowing us to make conclusions. When considering biomedical research, on average the target population will be 50% male and 50% female and we are now understanding that the differences between male and female are significant and stretch beyond hormonal differences. Therefore, as a general rule, including both sexes should be the first variable to be included to significant increase generalizability. However, there may be exceptions, such as the study of sex specific conditions or phenomena, as written into the NIH guidelines.”

– Please acknowledge that recruiting a sample that consists of 50% females and 50% males may increase variability (in some circumstances) compared to studies with 100% of one sex, and therefore change power calculations and/or require larger sample size.

The data is not pooled and hence the sex differences are accounted for. With a regression analysis, when the treatment effect is the same or in the same direction then there is no loss in power to estimating the average effect. We gain then that the conclusions space is larger. When the treatment effect is crossed, then we do lose power on estimating the average effect but we gain in ability to test for an interaction. However, in this situation if you had studied one sex you would have missed a major differences in the understanding of the outcome. We have reworded the section on page 8 to improve the clarity of this discussion.

9. Recommendations: Please consider discussing whether one option for closing the gap would be to fund/conduct studies that only include females.

We do not think that the sex data gap can be closed if data are collected in a sex-selective fashion. If generalizable data are to be collected, then biasing the sample in the opposite direction is not helpful. If the suggestion is to fund data collection including only females post-hoc, then it is not clear what justifies data collection of the first sample with only males. The only way to resolve this is to collect data on males and females from the get-go.

10. Line 315: Please provide a copy of the workshop program booklet or guide as supplemental information, so that "the way [you] constructed these workshops and facilitated discussion around sex could be used as a template for exploring other aspects of difference in future as research progresses". The reviewer was unable to find this information based on the references provided [75-77].

We have now included a copy of the workshop program in the supplemental information on the Open Science Framework page. We have also expanded our description and links in the text:

“In a three-part virtual workshop (June 2020). we explored practices, barriers, and challenges in designing and executing inclusive research in circadian physiology and sleep research. […] Across the three workshops, between 38 and 94 attendees participated in the interactive sessions, with approximately 4/5 of all participants being researchers (82/94 in Workshop 1, 47/60 in Workshop 2, 31/38 in Workshop 3).”

In the workshop descriptions, to make it easier for the reader, we now explicitly write:

“The video recording for Workshop 1 is available at https://osf.io/4mk9g/. The materials related to the participatory part are available at https://osf.io/y3ha2/.”

“The video recording for Workshop 2 is available at https://osf.io/ep6uh/. The materials related to the participatory part are available at https://osf.io/d32qr/.”

“The video recording for Workshop 3 is available at https://osf.io/rtaqu/. The materials related to the participatory part are available at https://osf.io/ntjbv/.”

11. Line 433 onwards (Methods): Please report whether one or more authors conducted the bibliometric analyses. If multiple authors contributed, please indicate whether there were any inter-rater reliability tests done to assess accurate coding.

We have clarified that only one reviewer reviewed the articles.

12. Please provide the source data for the bibliometric analysis, which should contain citation data and sex-based coding as presented in Figure 1.

Source data are included in the supplementary material.